# NGSpop: A desktop software that supports population studies by identifying sequence variations from next-generation sequencing data

Dong-Jun Lee[1]*, Taesoo Kwon[2], Hye-Jin Lee[1], Yun-Ho Oh[1], Jin-Hyun Kim[1], Tae-Ho Lee[1]

1 Genomics Division, National Institute of Agricultural Science, Jeonju, Republic of Korea, 2 Corporate R&D Center, Cloud9, Cheongju-si, Republic of Korea

* leemoses1004@gmail.com

**Data Availability Statement:** The compiled binary and test datasets are available at https://sourceforge.net/projects/ngspop/.

## Abstract

Next-generation sequencing (NGS) is widely used in all areas of genetic research, such as genetic disease diagnosis and breeding, and it can produce massive amounts of data. The identification of sequence variants is an important step when processing large NGS datasets; however, currently, the process is complicated, repetitive, and requires concentration, which can be taxing on the researcher. Therefore, to support researchers who are not familiar enough with bioinformatics to identify sequence variations regularly from large datasets, we have developed a fully automated desktop software, NGSpop. NGSpop includes functionalities for all the variant calling and visualization procedures used when processing NGS data, such as quality control, mapping, filtering details, and variant calling. In the variant calling step, the user can select the GATK or DeepVariant algorithm for variant calling. These algorithms can be executed using pre-set pipelines and options or customized with the user-specified options. NGSpop is implemented using JavaFX (version 1.8) and can thus be run on Unix-like operating systems such as Ubuntu Linux (version 16.04, 18.0.4). Although several pipelines and visualization tools are available for NGS data analysis, most integrated environments do not support batch processes; thus, variant detection cannot be automated for population-level studies. The NGSpop software developed in this study has an easy-to-use interface and helps in rapid analysis of multiple NGS data from population studies. According to a benchmark test, it effectively reduced the carbon footprint in bioinformatics analysis by expending the least central processing unit heat and power. Additionally, this software makes it possible to use the GATK and DeepVariant algorithms more flexibly and efficiently than other programs by allowing users to choose between the algorithms. As a limitation, NGSpop currently supports only the sequencing reads in fastq format produced by the Illumina platform. NGSpop is freely available at https://sourceforge.net/projects/ngspop/.

**Funding:** This study was conducted with support from the Research Program for Agricultural Science & Technology Development (Project No. PJ01427601) of the National Institute of Agricultural Science (http://www.naas.go.kr/english/), Rural Development Administration, Republic of Korea. The funders had no role in study design, data collection and analysis, decision to publish, or preparation of the manuscript.

**Competing interests:** The authors have declared that no competing interests exist.

## Introduction

Next-generation sequencing (NGS) is widely used in all areas of genetic research, such as disease diagnosis and breeding, partly because it is a useful tool for the detection of sequence variations [1–3]. NGS technology was originally used to study individuals and small samples, but more recently, it has been used to study cohort-level populations. A medical study by the Undiagnosed Diseases Network (UDN) showed that a genetic diagnosis with NGS is valid, even if the disease is undiagnosed [4]. According to NGS, 21% of patients had changes in therapy, 37% in diagnostic testing, and 36% in variant-specific genetic counseling. NGS has also been used to construct an ultra-high-density genetic map for the identification of molecular markers for agricultural research [5, 6]. The research showed that genetic breeding with NGS is a valid and reliable tool to develop useful characters. NGS produces a large amount of data, especially for studies involving genetic diseases and breeding at the population level. The identification of sequence variants in these large datasets is one of the most important processing steps; however, currently, sequence variation detection is both complicated and repetitive. Genomics consortia, such as the 1000 Genomes Project [7], provide shell scripts that implement a standard operation procedure (SOP) for variant detection, which helps to standardize the process (https://github.com/ekg/1000G-integration). However, most of the SOP shell scripts in use are difficult to understand and automate. There are several workflows and tools available that facilitate quality control (QC), mapping, calling, annotation, and visualization of variations. Some tools have too many functions, and consequently, they can be difficult to learn and often require official training. Furthermore, for some tools, the lack of tool integration and the multiple options included in their functionality can confuse the user; additionally, choosing a suitable option can be time consuming. Many pipelines and workflows have been developed by commercial and open-source communities to support NGS data analysis. Pipelines such as the ngs_backbone [8] and GATK [9] provide simple commands to perform a complete NGS data analysis. Most pipelines offer only a command-line interface; consequently, the user needs to be trained in Unix/Linux commands, shell scripts, or Python. It is difficult to automate variant detection in population-level studies. Galaxy [10] and the CLC Genomics Workbench [11] provide users with easy-to-use graphical user interfaces (GUIs). Although there are many pipelines and integrated environments for NGS data analysis, each has its own strengths and limitations (Table 1).

For example, Annovar is a widely used tool that provides only the annotation function for variant data. Likewise, Ngs_backbone [8] is an easy-to-use pipeline; however, it supports only

**Table 1. Comparison of the user-friendly graphical interfaces and functions of the single nucleotide polymorphism (SNP) analysis pipelines.**

| Name<br>Analysis | Annovar | Ngs_backbone | inGAP | Galaxy | CLC Genomics Workbench | NGSpop |
|---|---|---|---|---|---|---|
| Quality Control (QC) | | O | O | O | O | O |
| Read Mapping | | O | O | O | O | O |
| Variant calling (GATK) | | O | O* | O | O* | O |
| Variant calling (DeepVariant) | | | | | | O |
| Variant annotation | O | | | O | O | O |
| Visualization | | | O | | | O |
| Manual mode | O | O | O | | O | O |
| Batch mode | | | | O | O | O |
| Multi-sample | O | | | | O | O |

* inGAP/CLC Genomics Workbench use its own algorithm for variant calling.

GATK for variant calls and does not offer any function for visualizing variant information. inGap facilitates various functions from read mapping to variant calling and visualization but does not support multi-sample analysis and is no longer being developed. Galaxy also provides all functions for variant calling from NGS data but does not support multi-sample analysis. Above all, the system administrator must enable a very difficult setting before it can be used since it is executed as a web service. The CLC Genomics Workbench supports multi-sample analysis while providing all functions for NGS data analysis. However, because it uses its own algorithm for variant calling, it is not compatible with variant call results using GATK or DeepVariant. Unlike the aforementioned tools, NGSpop provides QC for NGS data analysis as well as variant calling and visualization and supports multi-sample data analysis. To support sequence variation detection in population-level genomic studies, we developed NGSpop. This software accepts multiple NGS datasets and allows the user to select between the GATK or DeepVariant [12] calling algorithms. The functionalities for variant detection include QC, mapping, filtering, variant calling, and visualization. Moreover, NGSpop has two modes of action: a one-step mode that supports batch identification of variants and a step-by-step mode in which the user can verify the result of each step. When the user selects the one-step mode, NGSpop can be executed using pre-set options to exclude the time-consuming steps. NGSpop can only be used with Linux operating systems.

## Implementation

NGSpop was implemented using JavaFX (version 1.8), and the tools employed within it were compiled on Ubuntu Linux (version 18.0.4). The GNU compiler collection version 7.2.0, for Ubuntu Linux, was used as a C-language compiler.

## Materials and methods

### Tools used in the pipeline

The tools included in NGSpop were carefully chosen according to the pipeline of the National Agricultural Biotechnology Information Center (NABIC, Republic of Korea; Fig 1). NGS data need to be evaluated for QC, and for this purpose, NGSpop includes FastQC (version 0.11.5). Filtering and trimming of the NGS data are mandatory, depending on the sequence quality, and for this step, NGSpop employs TrimmOmatic (version 0.36) [13]. After the QC step,

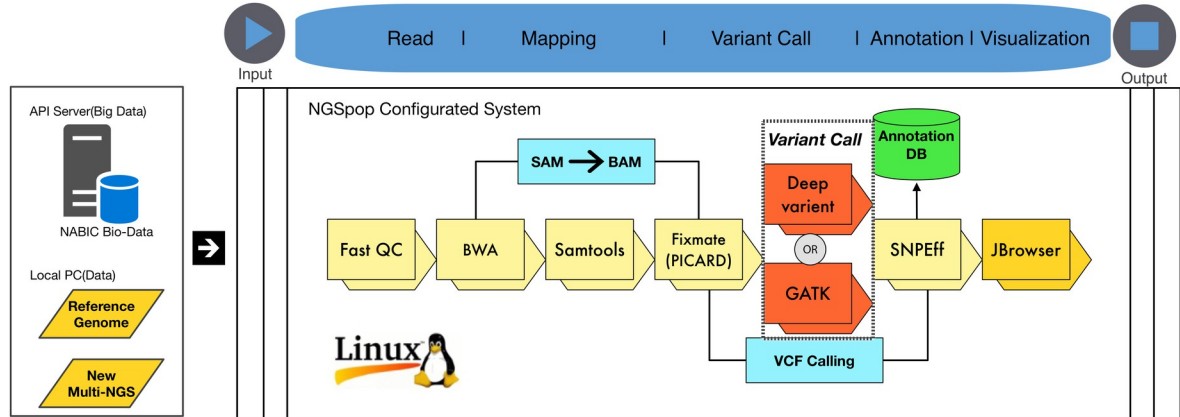

**Fig 1. NGS data analysis pipeline used in the NGSpop software.** The variant analysis protocol and tools are chosen according to the pipeline of the National Agricultural Biotechnology Information Center (NABIC, Republic of Korea).

**Table 2. Tools included in the NGSpop software.**

| Step | Tool | Version | Reference |
|---|---|---|---|
| QC | FastQC | 0.11.5 | (https://www.bioinformatics.babraham.ac.uk/projects/fastqc/) |
| | TrimmOmatic | 0.36 | [13] |
| Alignment | BWA | 0.7.16a | [14] |
| Post-processing | Samtools | 0.1.18 | [15] |
| | Picard | 2.9.4 | (http://broadinstitute.github.io/picard/) |
| | BamTools | 2.4.2 | [18] |
| | GATK (IndelRealigner) | 3.7.0 | [9] |
| Variant calling | GATK (HaplotypeCaller) | 3.7.0 | [9] |
| | GATK (UnifiedGenotyper) | 3.7.0 | [9] |
| | DeepVariant | 0.5.1 | [12] |
| Variant annotation | SnpEff | 4.3q | [16] |
| Visualization | Jbrowser | 1.12.3 | [17] |

The tools are listed in the order of their use in the pipeline.

sequence reads can be mapped in NGSpop against a reference genome using an alignment tool, such as BWA (version 0.7.16a) [14], and SAMtools [15] is used for file format conversion and indexing. Mate-pair information cannot be concordant with the sample library information and should be fixed. If sequence reads can be mapped to more than two loci, then the duplicate reads should be removed, and Picard (version 2.9.4) is used for this in NGSpop. For single nucleotide polymorphism (SNP)/INDEL identification, the user can select SNP/INDEL identification algorithms from the Genome Analysis Toolkit (version 3.7.0) or DeepVariant (version 0.5.1). Currently, DeepVariant is only supported by the Linux operating system; consequently, Linux is required to run NGSpop. To annotate the identified SNP/INDELs, SnpEff is used (version 4.3q) [16]. The identified and annotated variants are visualized using Jbrowser software (version 1.12.3) [17]. All the tools integrated into NGSpop are summarized in Table 2.

## Project creation and importing input files

The user must create a project and specify the data files, including fastq files of sequencing reads and a reference file in the FASTA format (Suppl. 2a). Fastq files of the sequencing reads can be multiple pairs of forward and reverse reads for population studies. Only fastq files produced by the Illumina platform can be processed using NGSpop. For the convenience of users, NGSpop can download a reference file from a genomic database such as the National Center for Biotechnology Information (NCBI), Ensemble, and the NABIC server through the application program interface. When the index file of the reference sequence does not exist, NGSpop performs indexing of the reference file.

## Step-by-step mode

NGSpop provides a step-by-step mode to expert users, in which they can investigate each step of the analysis. The user can change or execute each option during each step, and the changes will become the default options for the same step in each subsequent run (Suppl. 2b). Using the step-by-step mode, the user can finetune each option of the third-party tools used in every step. In addition, NGSpop provides the user with a log window to monitor the progress of each step.

## One-step mode

To automate NGS data analysis and support the large-scale identification of variants, NGSpop provides a one-step user mode that can run all processes employed by NGSpop with a single click. One-step mode is suitable for beginners who do not want to select complex options and for batch jobs. When NGSpop runs using the one-step mode, the default options will be used for each step. The user can customize the default options used in the one-step mode by first changing them using the step-by-step mode.

## Read mapping and duplicate removal

NGSpop employs BWA (version 0.7.16a) [14] as a mapping tool for the NGS reads. To convert the BWA sequence alignment map format (sam) to a binary alignment map (bam) format, and then sort and index the file, NGSpop uses SAMtools [15]. If the mate-pair information is not concordant with the sample library information, it should be verified and fixed. For this purpose, the Fixmate command of Picard (version 2.9.4) is used in NGSpop. In addition, duplicate reads are removed using the MarkDuplicates and AddOrReplaceReadGroups commands of Picard, and to calculate the statistics of the sequence reads, BamTools [18] is used.

## SNP/INDEL identification

Using NGSpop, the user can select a SNP/INDEL identification algorithm from the Genome Analysis Toolkit (version 3.7.0) or DeepVariant (version 0.5.1). The Genome Analysis Toolkit (version 3.7.0) [9] is a standard tool for SNP/INDEL identification from NGS data. To realign the reads around the INDELs, NGSpop uses the RealignerTargetCreator and IndelRealigner commands of GATK. After the realignment of the reads, UnifiedGenotyper is used as a variant caller in NGSpop.

## Variant annotation

Variations in the nucleotides can change the amino acids of the genes and thus affect the organism. Therefore, the functional effects of these variants on the genes should be predicted. To annotate the identified variants, NGSpop uses SnpEff (version 4.3q) [16]. In this study, NGSpop included only the *Arabidopsis thaliana* database (TAIR10 genome [19]) for SnpEff. In other studies, the SnpEff database should be included for the appropriate organism if it is available. If there is no available database for non-model organisms, the database should be generated manually.

## Software benchmarking

To conduct the benchmark test, we generated five test datasets using the complete *A. thaliana* genome sequencing data from the DNA Data Bank of Japan (DDBJ) FTP site under the accession number SRR519473 (paired-end run with 52,154,720 reads and 10,430,944,000 bp) [20]. The sequencing data were generated by the 1001 Genomes Project for *A. thaliana* (http://1001genomes.org) using the Illumina HiSeq 2000 platform. The dataset was mapped to the reference genome sequence of *A. thaliana* (Accession: GCF_000001735.4, TAIR version 10) that was downloaded from The Arabidopsis Information Resource (TAIR) FTP site. To show the general usability of the software, we performed another benchmark test on human exome sequencing data. We retrieved the whole-genome sequencing data of NA12878 under the accession number SRR003293 (8,843,809 reads) from the 1000 Genomes Project consortium ftp site (ftp://ftp.1000genomes.ebi.ac.uk/vol1/ftp/) [7]. To calculate the accuracy of the variant call result, it was compared with the dbSNP data. dbSNP data was downloaded from NCBI in

vcf format and version 32 was used. To compare dbSNP and variant call results, vcf-compare of vctools [21] was used. We compared the results obtained using NGSpop with those obtained using the ngs_backbone pipeline and CLC Genomics Workbench on a Xeon server. Detailed specifications of the Xeon server are as follows: Intel® X®(R) central processing unit (CPU) E5-2609 v3 @ 1.90 GHz, 32 GB memory, 220 GB solid-state drive for the booting device, 1.8 TB hard disk drive (HDD), and Ubuntu 18.0.4.

## Computational efficiency and energy consumption

To measure computation efficiency during the benchmarking test, we measured the CPU temperature. To measure the CPU temperature, logs were recorded using the psensor tool (https://github.com/chinf/psensor) and analyzed during the benchmark test. To measure the amount of electricity expended by the workstation performing the benchmark, hardware that can measure the amount of electricity by being plugged into a power plug was used.

## Results and discussion

The aim of NGSpop is to provide users with an easy-to-use environment for NGS data analysis, regardless of whether the user is an expert in bioinformatics. To this end, NGSpop provides users with two modes: a step-by-step mode for experts and a one-step mode for beginners. There are currently many workflows and easy-to-use tools available for NGS analysis, but as the user is required to run each step manually and wait until each step ends before proceeding, the rate of analysis is slow. Furthermore, these tools were not designed for population studies, and they only provide users with a step-by-step mode or a difficult hierarchical workflow design. Some tools do provide user-bash script interfaces, but these can be difficult to learn. However, when using NGSpop, only a single click of the run button is necessary and the results can be visualized using JBrowser (Fig 2). Even though the software provides a user-friendly GUI, NGSpop accepts multiple pairs of fastq files to support population-level studies. Thus, users can identify variants in large scale datasets from population studies using only their personal computers (PCs) or workstations. Moreover, NGSpop provides a selection of variant calling algorithms from GATK and DeepVariant in the variant calling steps. Variant calling is an important step in NGS data analysis and genetic studies. There are many tools that identify high-quality and reliable variants from NGS data, but none of them can identify all variants. Therefore, researchers have used and combined multiple tools to identify variants from NGS data.

## Software benchmarking

**System performance and efficiency.**   Recently, various disasters caused by global warming are causing great damage worldwide. Consequently, carbon emissions must be reduced to slow global warming. For example, when record temperatures hit much of the United Kingdom in July 2022, Google and Oracle suffered outages as cooling systems failed at London data centers (https://www.bbc.com/news/technology-62202125). Due to the effects of global warming, it has become crucial to consider energy efficiency and carbon footprint in bioinformatics analysis. Recently, Grealey et al. calculated the carbon footprint of analysis for each bioinformatics research field, such as genome-wide association studies, RNA sequencing, genome assembly, metagenomics, phylogenetics, and molecular simulations [22]. In the current study, we also compared the computational efficiency and energy consumption of the tools to determine how NGSpop contributes to the carbon footprint (See Materials and Methods).

**Running time.**   The running times of each tool investigated in the benchmark test are shown in Fig 3. NGSpop took an average of 19 min 26 s from processing the raw reads to

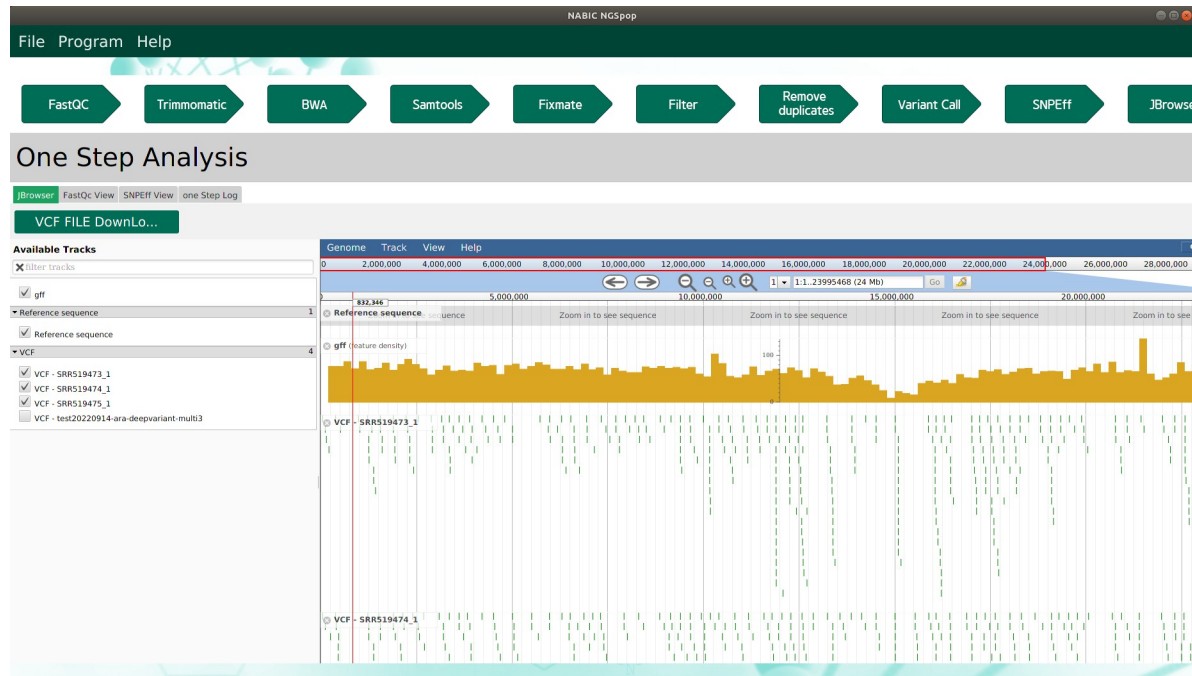

**Fig 2. Visualization of variants.** Four feature tracks are listed on the left panel of the JBrowser: reference sequence, annotation information of reference in GFF, mapped reads, and annotated variants. Only a.vcf file can be displayed in the JBrowser when multiple NGS data are selected for variant analysis after the merging of multiple.vcf files.

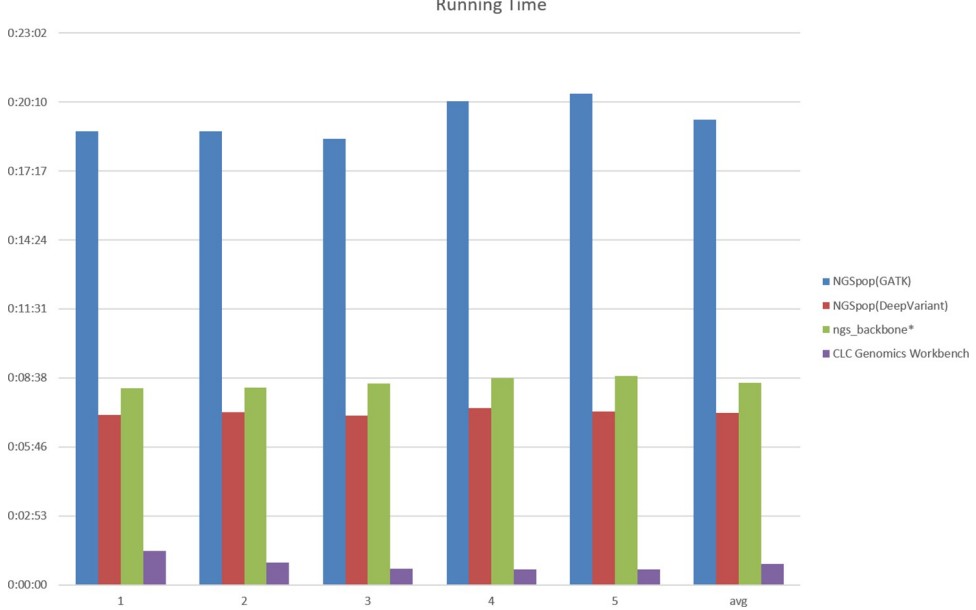

**Fig 3. Comparison of running times.** The running times of NGSpop, ngs_backbone, and CLC Genomics Workbench were compared. CLC Genomics Workbench showed the fastest performance, while NGSpop showed the slowest running time when using GATK algorithms. Because ngs_backbone does not have a visualization function, the visualization time of this software was excluded from the benchmark test.

annotating or visualizing the variants for the five test datasets using GATK, whereas it took 7 min 12 s when using DeepVariant. In contrast, ngs_backbone took an average of 8 min 27 s from processing the raw reads to annotating the variants. The total time required for the CLC genomics workbench to perform and visualize the variant call was 52 s, showing the fastest and best performance. NGSpop showed the slowest execution speed when using the GATK algorithm and the fastest execution speed among freeware when using the DeepVariant algorithm. For quick variant calling, it is better to use NGSpop with DeepVariant algorithms. In contrast, CLC Genomics Workbench demonstrated the fastest performance among the programs that underwent the benchmark test, which is presumed to be because its algorithm had been developed and optimized for a long time.

**CPU temperature.** We measured the temperature of the CPU to evaluate the computational efficiency, which affects the carbon footprint. Analysis of NGS data in a small capacity on a PC does not generate a lot of heat. In contrast, analysis of NGS data in a population unit that uses a workstation or high-performance computer will generate a lot of heat. Additionally, in order to lower the heat, considerable amounts of electricity must be used for cooling and ventilating the systems. Thus, a small difference in CPU heat can make a huge difference in carbon footprint. According to the benchmark test, the three programs did not show a significant difference in CPU heat (Fig 4). However, a large difference between the CPU heat generated when using NGSpop with DeepVariant algorithm, which was relatively low, and that generated with other algorithms was observed. Therefore, when analyzing large-capacity NGS data in a population unit, the use of NGSpop with DeepVariant algorithm may reduce the carbon footprint and be environmentally friendly.

**Power consumption.** To evaluate the power consumption, we used a hardware that measures the amount of electricity expended by being plugged into a power plug. Thus, the amount of power actually used was measured. The difference in the power consumption of

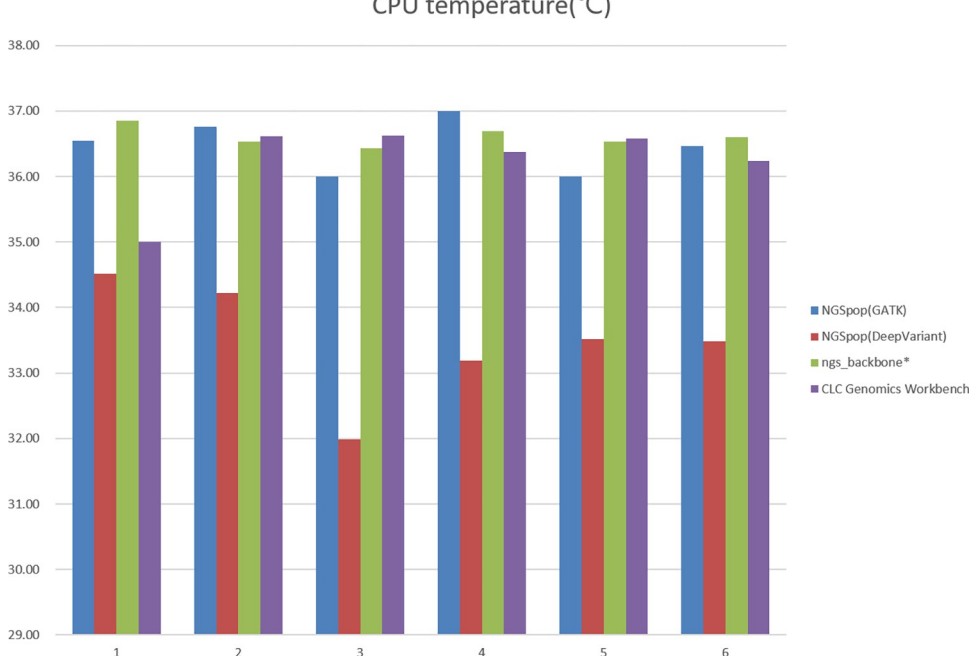

**Fig 4. CPU temperatures during the benchmark test.** CPU temperatures were measured for NGSpop, ngs_backbone, and CLC Genomics Workbench during the benchmark test. Although the three programs showed no significant difference, CPU heat was greatly reduced when using NGSpop with DeepVariant algorithm.

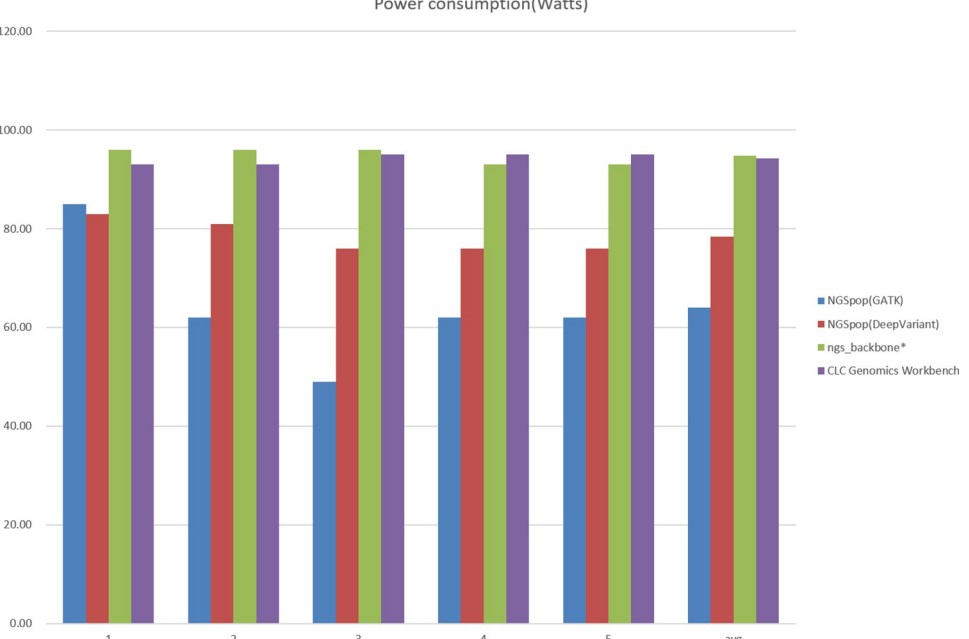

**Fig 5. Comparison of power consumption.** Power consumption of the workstation was measured for NGSpop, ngs_backbone, and CLC Genomics Workbench during the benchmark test. When NGSpop with GATK algorithm was used, the lowest amount of power was used.

ngs_backbone and CLC Genomics Workbench was not large. However, NGSpop used relatively little power and showed a larger difference compared to the two programs (Fig 5). In particular, using the NGSpop with the GATK algorithm used less power than with the DeepVariant algorithm. Therefore, if one wishes to analyze NGS data using less power, it is better to use NGSpop with the GATK algorithm.

**Comprehensive consideration.** According to the benchmark test, CLC Genomics Workbench showed the fastest performance in running time whereas NGSpop was the slowest when using the GATK algorithm. Nevertheless, NGSpop showed better performance than the other two programs in terms of CPU heat and power consumption. In addition, when NGSpop using DeepVariant algorithm was used, the data analysis speed was faster than when ngs_backbone was used; therefore, it was possible to compensate for the disadvantages of the GATK algorithm. Although NGSpop does not surpass the latest commercial program, CLC Genomics Workbench, it can be used more flexibly and efficiently than other freeware by offering users the option to choose from the GATK and DeepVariant algorithms. Additionally, it reduces the carbon footprint of bioinformatics analysis by reducing CPU heat and power consumption. Thus, NGSpop was found to be an easy-to-use platform for variant calling and was efficient in computation and power consumption.

## Conclusions

Large-scale parallel sequencing has become a popular tool to identify sequence variations, and many tools have now been developed to analyze NGS data. Although many tools have been developed, few support population-level or cohort-level sequencing data. Owing to the lack of population-level analysis tools, many researchers find it difficult to analyze the massive volumes of NGS data they produce. Researchers should ideally write scripts to analyze NGS data

on the Linux command line. NGSpop is a user-friendly software for researchers who are not familiar with the command line interface and do not want to write shell scripts. Therefore, NGSpop provides the user with an easy-to-use interface and helps to automate the detection of variations from the NGS data at the population level. It helps genomics researchers who want to analyze population-level NGS data with an easy-to-use GUI. The results of the benchmark test indicated that NGSpop used with the GATK algorithm was the slowest program. Nevertheless, it effectively reduced the carbon footprint in bioinformatics analysis by expending the least CPU heat and consuming the least power. In addition, it was possible to use the GATK and DeepVariant algorithms more flexibly and efficiently than other programs by offering users the option to choose between the algorithms. However, there are some limitations. First of all, NGSpop only accepts fastq format data that has been produced using Illumina platform because there are too many parameters to consider when analyzing all types of NGS platforms. Next, NGSpop can only be supported by the Linux operating system because DeepVariant, one of the variant calling algorithms used by this program can only be used with Linux operating systems. Moreover, in order to install NGSpop, the user must have the ability to install the perl module and some third-party programs using Linux. Therefore, we intend to provide a virtual machine image of Oracle's VirtualBox pre-installed with NGSpop and third-party programs in the future. We will also include functionalities that support NGS platforms other than Illumina while accounting for the variations in formats in further studies.

## Supporting information

**S1 File.**
(DOCX)

## Author Contributions

**Conceptualization:** Dong-Jun Lee, Taesoo Kwon, Tae-Ho Lee.

**Data curation:** Dong-Jun Lee, Taesoo Kwon, Hye-Jin Lee, Yun-Ho Oh, Jin-Hyun Kim.

**Formal analysis:** Dong-Jun Lee, Taesoo Kwon.

**Funding acquisition:** Dong-Jun Lee, Tae-Ho Lee.

**Investigation:** Dong-Jun Lee.

**Methodology:** Dong-Jun Lee, Taesoo Kwon, Tae-Ho Lee.

**Project administration:** Dong-Jun Lee, Taesoo Kwon.

**Resources:** Dong-Jun Lee, Taesoo Kwon.

**Software:** Dong-Jun Lee, Taesoo Kwon, Tae-Ho Lee.

**Supervision:** Dong-Jun Lee.

**Validation:** Dong-Jun Lee, Taesoo Kwon.

**Visualization:** Dong-Jun Lee, Taesoo Kwon.

**Writing – original draft:** Dong-Jun Lee, Taesoo Kwon.

**Writing – review & editing:** Dong-Jun Lee, Taesoo Kwon, Hye-Jin Lee, Yun-Ho Oh, Jin-Hyun Kim, Tae-Ho Lee.

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
