## [Decision Letter · Decision Letter 0]

7 Apr 2022

PONE-D-21-36496NGSpop: A desktop software that supports population studies by identifying sequence variations from next-generation sequencing dataPLOS ONE

Dear Dr. Lee,

Thank you for submitting your manuscript to PLOS ONE. After careful consideration, we feel that it has merit but does not fully meet PLOS ONE’s publication criteria as it currently stands. Therefore, we invite you to submit a revised version of the manuscript that addresses the points raised during the review process.

We look forward to receiving your revised manuscript.

Kind regards,

Mehdi Rahimi, Ph.D.

Academic Editor

PLOS ONE

Journal Requirements:

Reviewers' comments:

Reviewer's Responses to Questions

**Comments to the Author**

1. Is the manuscript technically sound, and do the data support the conclusions?

Reviewer #1: Yes

Reviewer #2: Partly

Reviewer #3: Partly

2. Has the statistical analysis been performed appropriately and rigorously? 

Reviewer #1: N/A

Reviewer #2: N/A

Reviewer #3: N/A

3. Have the authors made all data underlying the findings in their manuscript fully available?

Reviewer #1: Yes

Reviewer #2: Yes

Reviewer #3: Yes

4. Is the manuscript presented in an intelligible fashion and written in standard English?

Reviewer #1: Yes

Reviewer #2: No

Reviewer #3: Yes

5. Review Comments to the Author

Reviewer #1: NGSpop implements a read mapping/variant calling pipeline. NGSpop is based on JavaFX. The paper is well-written and easy to follow.

Major comments:

1. As the authors acknowledge, there are already many NGS pipelines. NGSpop is positioned as one of the few pipelines that supports handling many samples. When processing many samples, the issue of computational efficiency/energy efficiency becomes important. The manuscript does not really deal with this aspect. However, it feel important and maybe, the authors want to discuss this as well.

2. The only comparison to state-of-the-art tools is provided in Table 1, in the form a listing of features for individual tools. Yet, not actual comparison is performed. Providing a more detailed comparison against one or two state-of-the-art tools in terms of functionality and/or performance and/or accuracy of the obtained results would be informative.

3. It seems strange that the benchmark test is performed on Arabidopsis, rather than on human. This makes it very difficult to compare runtime results against benchmark results provided in other papers (often performed on human). Is there a specific reason why benchmarks were done on Arabidopsis?

Reviewer #2: The authors provide details of a pipeline for NGS analysis using a set of third-party tools. They acknowledge that many users will want to define their own pipeline and run their own analysis using the command line, NGSPop is not set to replace this but provide a UI and user-friendly method of launching third party tools.

In essence, it appears to be a front end to other tools and the novelty and originally is in the ability to provide access to the tools via a UI and not the command line.

There is little comment aside from Table 1, on other similar pipelines, Table 1 lists a number with user friendly graphical interfaces but there appears to be little further discussion of the pros and cons of these other pipelines. This would be of benefit to the reader in assessing the suitability of NGSPop

There is considerable content in the paper describing the third-party tools used at each stage. Is this necessary? Could this just be done in table 2.

It is unclear how the basic and expert modes differ, and this could be explored further in the paper. The benefits of the command line and bespoke pipelines are the ability to adjust parameters for the individual tools, in response to the data, rather than a one-size-fits all approach. Is this retained or is the one-click option restrictive to fine tuning the analysis. It is unclear from the paper, and I have not tested the tool directly, how much flexibility the user has in selecting options within the third-party tools.

It could also be argued that in using Linux and Perl (with sub-modules) some degree of familiarity with the command line is required, potentially during any installation of if issues arise with the Perl installations. It is also unclear if the user is expected to install any of the third-party tools required or if the installer for NGSPop completes this step. The Supplementary files details only the Perl and a single apt install command. This suggests the setup and installation requires a user skill level beyond that of the target audience. If this is the case would the user be more familiar with their own custom pipelines, rather than the offerings from NGSPop.

The figure quality was low in the proofs provided and should be checked at further stages in the process.

The authors should be commended for looking at providing a more user-friendly graphical interface to many tools, though the paper would benefit from discussions of how this works in practice, maybe with examples, and highlighting the pros and cons of NGSPop, and ease of use, rather than listing the third-party tools used.

Reviewer #3: The manuscript describes a software for next-generation sequencing (NGS) data analysis that support population studies. The software, named NGSpop is an assembly of several bioinformatc tools well-known in NGS data field, with the aim to simplifying the analytical pipeline, also to researchers who are not familiar with bioinformatics.

The manuscript provides details of the several tools which compose the NGSpop, with a description of the software functionality.

In the “Abstract” section should be stated that NGSpop can processed only sequenced reads in fastq format produced by Illumina platform. This is a an important info that must be immediately clear to the reader.

Moreover, it should be clearly stated if the software in freely available or not.

The “Results and discussion“ section doesn’t seem to make sense, but seems to be a kind of software validation, with a small set of NGS data. The text from row 204 to row 223 contains info already exhaustively explained in the previous sections, and this section could be implemented with additional details about the software validation.

6. PLOS authors have the option to publish the peer review history of their article (what does this mean?). If published, this will include your full peer review and any attached files.

Reviewer #1: **Yes: **Jan Fostier

Reviewer #2: No

Reviewer #3: No

---

## [Author Response · Author response to Decision Letter 0]

10 Oct 2022

Detailed Responses to Reviewers

September 15, 2022

PONE-D-21-36496

NGSpop: A desktop software that supports population studies by identifying sequence variations from next-generation sequencing data

PLOS ONE

Dear Editor: 

We have carefully considered all the comments by the reviewers and have revised the manuscript accordingly. Listed below are the comments and our point-by-point responses (blue text) to each comment, through which we detailed the changes made to the text and their locations in the revised manuscript. All page numbers refer to those in the revised text, which has been submitted online.

We hope that the revised manuscript is now acceptable for publication in your esteemed journal.

Yours sincerely,

Dong-Jun Lee

Reviewers' comments:

Reviewer's Responses to Questions

Comments to the Author

1. Is the manuscript technically sound, and do the data support the conclusions?

Reviewer #1: Yes

Reviewer #2: Partly

Reviewer #3: Partly

2. Has the statistical analysis been performed appropriately and rigorously?

Reviewer #1: N/A

Reviewer #2: N/A

Reviewer #3: N/A

3. Have the authors made all data underlying the findings in their manuscript fully available?

Reviewer #1: Yes

Reviewer #2: Yes

Reviewer #3: Yes

4. Is the manuscript presented in an intelligible fashion and written in standard English?

Reviewer #1: Yes

Reviewer #2: No

Reviewer #3: Yes

5. Review Comments to the Author

Reviewer #1: 

NGSpop implements a read mapping/variant calling pipeline. NGSpop is based on JavaFX. The paper is well-written and easy to follow.

Major comments:

1. As the authors acknowledge, there are already many NGS pipelines. NGSpop is positioned as one of the few pipelines that supports handling many samples. When processing many samples, the issue of computational efficiency/energy efficiency becomes important. The manuscript does not really deal with this aspect. However, it feel important and maybe, the authors want to discuss this as well.

Response: We thank the reviewer for this suggestion. Accordingly, we have compared the computational efficiency and energy efficiency of NGSpop with that of other similar programs in the study. We measured the temperature of the CPU using a system management tool (psensor). In addition, the amount of power used during program execution was measured using a power measuring device. We have described these in the following sections:

- “Computational efficiency and energy consumption” in Materials and Methods (Lines 208 to 214)

- “CPU temperature,” “Power consumption,” and “Comprehensive consideration” in Results and Discussion (Lines 277 to 324).

2. The only comparison to state-of-the-art tools is provided in Table 1, in the form a listing of features for individual tools. Yet, not actual comparison is performed. Providing a more detailed comparison against one or two state-of-the-art tools in terms of functionality and/or performance and/or accuracy of the obtained results would be informative.

Response: We appreciate this comment. As mentioned herein, it is necessary to compare NGSpop with other tools to obtain detailed information to be provided to the users so that they can select the appropriate tool according to their intended purpose. We performed a benchmark test to compare the system performance and efficiency, running time, CPU temperature and power consumption. The results of comparison are described in the Results and Discussion (Lines 242 to 323; Figures 3 to 5).

3. It seems strange that the benchmark test is performed on Arabidopsis, rather than on human. This makes it very difficult to compare runtime results against benchmark results provided in other papers (often performed on human). Is there a specific reason why benchmarks were done on Arabidopsis?

Response: We appreciate this comment. Because NGSpop was developed for the analysis of the population genetics of plants, it was not analyzed using human data. Nevertheless, as suggested, we performed the benchmark test on human exome data and provided the results in the revised manuscript (Figures 3 to 5).

Reviewer #2: 

The authors provide details of a pipeline for NGS analysis using a set of third-party tools. They acknowledge that many users will want to define their own pipeline and run their own analysis using the command line, NGSPop is not set to replace this but provide a UI and user-friendly method of launching third party tools.

In essence, it appears to be a front end to other tools and the novelty and originally is in the ability to provide access to the tools via a UI and not the command line.

There is little comment aside from Table 1, on other similar pipelines, Table 1 lists a number with user friendly graphical interfaces but there appears to be little further discussion of the pros and cons of these other pipelines. This would be of benefit to the reader in assessing the suitability of NGSPop.

Response: We thank the reviewer for this suggestion. Accordingly, we have added more detailed descriptions of similar pipelines in the revised manuscript (Table 1, lines 78 to 90). We also compared the differences between the tools in terms of actual performance through benchmark tests (Lines 242 to 323; Figures 3 to 5).

There is considerable content in the paper describing the third-party tools used at each stage. Is this necessary? Could this just be done in table 2.

Response: We appreciate this comment from the reviewer. As suggested, we deleted repetitive descriptions for the third-party tools by deleting the “Quality control,” “DeepVariant,” “Variant merge,” and “Variant visualization” sections. However, we retained the sections titled “Read mapping and duplicate removal,” “SNP/INDEL identification,” and “Variant annotation” as a specific option of the third-party tool was explained therein.

It is unclear how the basic and expert modes differ, and this could be explored further in the paper. The benefits of the command line and bespoke pipelines are the ability to adjust parameters for the individual tools, in response to the data, rather than a one-size-fits all approach. Is this retained or is the one-click option restrictive to fine tuning the analysis. It is unclear from the paper, and I have not tested the tool directly, how much flexibility the user has in selecting options within the third-party tools.

Response: We appreciate this thoughtful comment. The benefits of the command line tool are retained in NGSpop through the step-by-step mode (expert mode). If an expert user changes the default options using Step-by-step mode, the same changes will be automatically implemented in the one-step mode. We have added more detailed descriptions of the two modes in the sections titled “Step-by-step mode” and “One-step mode” under Materials and Methods.

It could also be argued that in using Linux and Perl (with sub-modules) some degree of familiarity with the command line is required, potentially during any installation of if issues arise with the Perl installations. It is also unclear if the user is expected to install any of the third-party tools required or if the installer for NGSPop completes this step. The Supplementary files details only the Perl and a single apt install command. This suggests the setup and installation requires a user skill level beyond that of the target audience. If this is the case would the user be more familiar with their own custom pipelines, rather than the offerings from NGSPop.

Response: We appreciate this thoughtful comment. We developed this program by keeping convenience of usage in mind but did not consider the convenience of installation. Therefore, it must be installed by a Linux system administrator. We have mentioned this as one of the shortcomings of this program in the Conclusions section.

In addition, it is sometimes difficult to install Perl modules in the Linux operating system. Therefore, we intend to provide NGSPop pre-installed in an Ubuntu system as a virtual machine image through VirtualBox (Oracle Inc.) in the future.

The figure quality was low in the proofs provided and should be checked at further stages in the process.

Response: We appreciate this comment. As suggested, we have created high-quality figures according to the journal specifications and submitted them with the revised manuscript.

The authors should be commended for looking at providing a more user-friendly graphical interface to many tools, though the paper would benefit from discussions of how this works in practice, maybe with examples, and highlighting the pros and cons of NGSPop, and ease of use, rather than listing the third-party tools used.

Response: We thank the reviewer for this suggestion. As mentioned, it is necessary to compare NGSpop with other tools to provide detailed information so that users can select the tool most suited to their purpose. We performed a benchmark test to compare the system performance and efficiency, running time, CPU temperature and power consumption. The results of comparison are described in the Results and Discussion (Lines 242 to 323; Figures 3 to 5).

Reviewer #3: The manuscript describes a software for next-generation sequencing (NGS) data analysis that support population studies. The software, named NGSpop is an assembly of several bioinformatc tools well-known in NGS data field, with the aim to simplifying the analytical pipeline, also to researchers who are not familiar with bioinformatics.

The manuscript provides details of the several tools which compose the NGSpop, with a description of the software functionality.

In the “Abstract” section should be stated that NGSpop can processed only sequenced reads in fastq format produced by Illumina platform. This is an important info that must be immediately clear to the reader.

Response: We appreciate this comment. As suggested, we have stated the limitation of NGSPop software in the Abstract section.

Moreover, it should be clearly stated if the software in freely available or not.

Response: We appreciate this helpful comment. As suggested, we have mentioned in the Abstract section that NGSpop is freely available.

The “Results and discussion“ section doesn’t seem to make sense, but seems to be a kind of software validation, with a small set of NGS data. The text from row 204 to row 223 contains info already exhaustively explained in the previous sections, and this section could be implemented with additional details about the software validation.

Response: We appreciate this comment from the reviewer. As mentioned, it is necessary to compare NGSpop with other tools to provide detailed information so that users can select the tool most suited to their purpose. We performed a benchmark test to compare the system performance and efficiency, running time, CPU temperature and power consumption and described the results of the comparison in the Results and Discussion as well as the Conclusions.

6. PLOS authors have the option to publish the peer review history of their article (what does this mean?). If published, this will include your full peer review and any attached files.

Do you want your identity to be public for this peer review? For information about this choice, including consent withdrawal, please see our Privacy Policy.

Reviewer #1: Yes: Jan Fostier

Reviewer #2: No

Reviewer #3: No

---

## [Decision Letter · Decision Letter 1]

31 Oct 2022

NGSpop: A desktop software that supports population studies by identifying sequence variations from next-generation sequencing data

PONE-D-21-36496R1

Dear Dr. Lee,

We’re pleased to inform you that your manuscript has been judged scientifically suitable for publication and will be formally accepted for publication once it meets all outstanding technical requirements.

Kind regards,

Mehdi Rahimi, Ph.D.

Academic Editor

PLOS ONE

Additional Editor Comments (optional):

Reviewers' comments:

Reviewer's Responses to Questions

**Comments to the Author**

1. If the authors have adequately addressed your comments raised in a previous round of review and you feel that this manuscript is now acceptable for publication, you may indicate that here to bypass the “Comments to the Author” section, enter your conflict of interest statement in the “Confidential to Editor” section, and submit your "Accept" recommendation.

Reviewer #1: All comments have been addressed

Reviewer #2: All comments have been addressed

Reviewer #3: All comments have been addressed

2. Is the manuscript technically sound, and do the data support the conclusions?

Reviewer #1: Yes

Reviewer #2: Yes

Reviewer #3: Yes

3. Has the statistical analysis been performed appropriately and rigorously? 

Reviewer #1: N/A

Reviewer #2: N/A

Reviewer #3: I Don't Know

4. Have the authors made all data underlying the findings in their manuscript fully available?

Reviewer #1: (No Response)

Reviewer #2: Yes

Reviewer #3: Yes

5. Is the manuscript presented in an intelligible fashion and written in standard English?

Reviewer #1: Yes

Reviewer #2: Yes

Reviewer #3: Yes

6. Review Comments to the Author

Reviewer #1: The authors have addressed my concerns. The results are what they are (cf. Fig. 3), however, potential users can now make a more informed decision on whether to adopt this tool.

Reviewer #2: (No Response)

Reviewer #3: (No Response)

7. PLOS authors have the option to publish the peer review history of their article (what does this mean?). If published, this will include your full peer review and any attached files.

Reviewer #1: No

Reviewer #2: No

Reviewer #3: No

---

## [Editor Report · Acceptance letter]

8 Nov 2022

PONE-D-21-36496R1 

NGSpop: A desktop software that supports population studies by identifying sequence variations from next-generation sequencing data 

Dear Dr. Lee:

I'm pleased to inform you that your manuscript has been deemed suitable for publication in PLOS ONE. Congratulations! Your manuscript is now with our production department. 

Kind regards, 

on behalf of

Dr. Mehdi Rahimi 

Academic Editor

PLOS ONE